# Semiconducting Properties of the Hybrid Film of Elastic Poly(styrene-*b*-butadiene-*b*-styrene) Block Copolymer and Semiconducting Poly(3-hexylthiophene) Nanofibers

**DOI:** 10.3390/polym12092118

**Published:** 2020-09-17

**Authors:** Takanori Goto, Jun Morita, Yuya Maekawa, Shinji Kanehashi, Takeshi Shimomura

**Affiliations:** Graduate School of Engineering, Tokyo University of Agriculture and Technology, Koganei, Tokyo 184-8588, Japan; goto-tkn@awi.co.jp (T.G.); Jun_Morita@jsr.co.jp (J.M.); s206536y@st.go.tuat.ac.jp (Y.M.); kanehasi@cc.tuat.ac.jp (S.K.)

**Keywords:** conducting polymer, composite, nanofiber

## Abstract

We investigated the electrical properties of a composite film loaded with semi-conductive poly(3-hexylthiophene) (P3HT) nanofibers dispersed in poly(styrene-*b*-butadiene-*b*-styrene) (SBS). This structure can be regarded as the hybrid of SBS matrix with elastic mechanical properties and P3HT nanofibers with semiconducting properties. The P3HT nanofibers were embedded in the fingerprint pattern of microphase-separated SBS, as observed by scanning force microscopy. Furthermore, the electrical conductivity and field-effect mobility of the composite films were evaluated. The field-effect mobility was estimated to be 6.96 × 10^−3^ cm^2^ V^−1^ s^−1^, which is consistent with the results of previous studies on P3HT nanofibers dispersed in an amorphous polymer matrix including poly(methyl methacrylate) and polystyrene, and we found that the P3HT nanofiber network was connected in the SBS bulk matrix. The film was stretchable; however, at elongation by two times, the nanofiber network could not follow the elongation of the SBS matrix, and the conductivity decreased drastically. The field-effect transistor of this film was operated by bending deformation with a radius of curvature of 1.75 cm, though we could not obtain an off-state and the device operated in a normally-on state.

## 1. Introduction

Nanofibers comprising conducting polymers have a quasi-one-dimensional structure, large specific surface area, flexible properties, and semiconducting properties; therefore, conducting polymer nanofibers are required for various applications, including molecular wires, polymer transistors, electrostatic shields, and biosensors [1,2,3,4,5]. There are various methods for fabricating conducting polymer nanofibers, particularly, the electrospinning method, which can be applied to various types of conducting polymers such as polyaniline [6,7,8,9], poly[2-methoxy-5-(2-ethylhexyloxy)-1,4-phenylenevinylene] [10], poly(3-hexylthiophene) (P3HT) [11], polypyrrole [12], poly(p-phenylenevinylene) [13], and poly(3,4-ethylenedioxythiophene):poly(styrene sulfonate) [14,15], has been widely used. Meanwhile, regioregular P3HT can be crystallized as whisker crystals with a fine structure of width and length of 15 nm and several micrometers, respectively [16], from supercooled solutions. This P3HT nanofiber is used as a p-type semiconductor in a field-effect transistor (FET) with a considerably high field-effect mobility, similar to those of P3HT thin films [17,18]. The FET characteristics of the P3HT nanofibers have been extensively investigated for thin-film states [19,20,21,22,23,24,25], entangled networks [26,27], and isolated single fibers [17,18,28,29].

Recently, it was reported that P3HT can be recrystallized as nanofibers in conventional noncrystalline polymers such as poly(methyl methacrylate) (PMMA) and polystyrene (PS) to form composite films loaded with P3HT nanofibers [30,31,32]. These composite films exhibit semiconducting properties and can be fabricated into flexible FETs. Using conductive atomic force microscopy and Kelvin probe force microscopy, an effective nanofiber network percolated over these films has been observed [33,34]. In addition, these composite films could be doped and represented as potential flexible films with considerable mechanical strength and high transparency, owing to the transparent matrix, and high conductivity, owing to the well-developed P3HT nanofiber network, with high environmental stability [35]. The FET characteristics of these films have also been investigated, and the rectifying property has been achieved by binding the composite loaded with n-type nanofibers [36]. From these results, it is expected that a P3HT nanofiber composite film can be regarded as a flexible semiconducting substrate that can substitute a nonflexible silicon substrate. However, the matrices of PMMA and PS used in the previous studies were rather brittle; therefore, these composite films loaded with the P3HT nanofibers were not flexible.

In this study, we used poly(styrene-*b*-butadiene-*b*-styrene) (SBS) copolymer as a flexible matrix and investigated the electrical properties of SBS composite films loaded with the P3HT nanofibers. The SBS matrix comprises a soft matrix of polybutadiene and hard microdomains of polystyrene and the solid microdomains act as physical cross-linkers in the soft matrix. This structure can be regarded as the hybrid of SBS matrix with elastic mechanical properties and P3HT nanofibers with semiconducting properties. Furthermore, as SBS is one of the suitable block copolymers for observing the phase structure by the scanning force microscope (SFM), we can confirm the formation and dispersion of P3HT nanofibers in SBS matrix by the SFM observation. It has been reported that SBS can be loaded with conductive fillers, such as Ag nanowires and carbon nanotubes (CNTs), to obtain elastic, flexible, and conductive films [37,38,39,40,41,42]. The formation of the P3HT nanofibers dispersed in a matrix with a crystal structure has not been reported so far. SBS has a microphase separation structure; hence, for the formation of conductive networks and good mechanical strength, it is important to investigate the relationship between the microdomains and nanofibers. Furthermore, we investigated the FET properties of the SBS flexible films loaded with the P3HT nanofibers. The novelty of this work is to develop the elastic and semiconducting polymer sheet using conducting polymers with a high elasticity as substitute for the rigid Si wafer. Although poly(3,4-ethyrene dioxythiophene):poly(styrenesulfonate) (PEDOT:PSS) is known to have high conductivity and applied to various applications, PEDOT:PSS is hydrophilic polymer which cannot be dispersed in almost elastomer with hydrophobic properties. The P3HT nanofiber is a realistic option for achieving the elastomer loaded with conducting polymers. There are some reports on this strategy by using polystyrene-*b*-poly(ethylene-*co*-butylene)-*b*-polystyrene [43,44,45], polydimethylsiloxane (PDMS) [46,47], and silicon rubber [48]. In this study, we particularly focused on the relationship between the structure of the microphase separation and P3HT nanofibers, which has not been discussed in the previous study.

## 2. Materials and Methods

Regioregular P3HT (Mw 44,000), SBS copolymer (Mw 140,000, styrene 30 wt%), and poly(methyl methacrylate) (PMMA) (Mw 120,000) were purchased from Sigma-Aldrich Co., Inc. (St. Louis, MO, USA) and used without further purification. Chloroform was purchased from FUJIFILM Wako Pure Chemical Co., Ltd. (Tokyo, Japan), and anisole and acetone were purchased from Kokusan Chemical Co., Ltd. (Tokyo, Japan).

The SBS composite films, in which the P3HT nanofibers were dispersed, were prepared by the following procedure for the preparation of PMMA and PS composite films [30]. The P3HT and SBS powders were added to a solvent mixture composed of chloroform and anisole and the solution was stirred for 60 min at 70 °C. Both the solvent mixture components, chloroform and anisole, are good and poor solvents for P3HT, respectively. The solution was prepared at a fixed P3HT weight ratio of 0.05 wt% and an SBS weight ratio of 0.5 wt% to 5 wt%, in a fixed 70:30 (*v*/*f*) solvent mixture of chloroform/anisole. The solution was cooled gradually to 20 °C at a rate of 15 °C/h without stirring. Further, this transparent yellow solution changed to a turbid reddish-brown suspension when it was left undisturbed in ambient air for one week. Thin films were prepared by spin-casting the suspension on a Si substrate with a thermally oxidized SiO_2_ layer (thickness = 255 nm) on top, and self-standing films were prepared by spin-casting the suspension on a glass substrate, followed by peeling it off of the substrate. For the FET measurements under bending deformation, we used a polypropylene (PP) sheet as a flexible substrate.

The surface observation of the composite thin films was performed using a scanning probe microscope (Nanocute/NanoNavi IIe, SII Nanotechnology Inc., Tokyo, Japan) in the SFM mode. The instrument was equipped with a commercial silicon cantilever (OMCL-AC160TS-C3, Olympus Corp., Tokyo, Japan) with a spring constant and a resonant frequency of approximately 26 N/m and 300 KHz, respectively. Conductivity measurements were performed using a two-probe method in a vacuum below 10^−5^ Torr using a source/measure unit (SMU; 6430, Keithley Instruments, Inc., Cleveland, OH, USA) in a cryogenic probing station (LIPS, Nagase Techno-Engineering Co. Ltd., Tokyo, Japan). Pt electrodes were sputter-deposited through a metal mask by reactive ion-beam etching (EIS-200ER, Elionix Inc., Tokyo, Japan), and the width *W* and gap *L* of the electrodes were 40 μm and 600 μm, respectively. The electrodes were deposited on the substrate in the case of the thin film, while they were deposited on the film in the case of the self-standing and flexible films for bending deformation.

The FET measurements were performed by the two-probe method in a vacuum below 10^−5^ Torr at 300 K using a system combining the cryogenic probing station LIPS, SMU 6430 to measure the source–drain characteristics, and a Keithley 2400 digital source meter (Keithley Instruments, Inc., Cleveland, OH, USA) for applying the gate field. For thin-film measurement, two electrodes on the substrate were used as the source and drain (bottom contact), and the backside of Si was used as a back gate. For FET measurements under the bending deformation, after sputter-deposition of the electrodes on the sample film (top contact), the PMMA solution in acetone (20 wt%) was spin-casted (1000 rpm, 90 s) as a gate insulator. Further, the gate electrodes were sputter-deposited using a metal mask as the top gate. The field-effect mobility, *μ*, was estimated using the following relationship:
(1)$$\mu =\frac{2L}{WC_{\mathrm{OX}}}{\left(\frac{\partial \sqrt{I_{\mathrm{SD}}}}{\partial V_{G}}\right)}^{2}$$
where *L* is the spacing between the electrodes, *W* is the width of the electrodes, and *C*_ox_ (=13.8 nm cm^−2^) is the capacitance of the insulation layer of SiO_2_ (255 nm thick). *I*_SD_ is the drain current and *V*_G_ is the gate voltage.

## 3. Results and Discussion

### 3.1. Microscopic Structure of the SBS Film Loaded with a P3HT Nanofiber

Figure 1a shows the SFM image of the thin film obtained by spin-casting the solution of SBS (0.5 wt%) in a mixture of chloroform and anisole (70:30 (*v*/*v*)). The typical fingerprint pattern associated with the microphase separation can be clearly observed, and the section plot (Figure 1b) shows that the domain spacing is approximately 50 nm.

Figure 2 shows the SFM images of the thin film obtained by spin-casting the suspension of P3HT (0.05 wt%) and SBS (0.5 wt%) in a mixture of chloroform and anisole (70:30 (*v*/*v*)). The fingerprint pattern of the microphase separation of SBS was observed in addition to the nanofiber formation of P3HT. From this result, it can be indicated that the SBS polymers in the suspension did not affect the P3HT nanofiber formation. Furthermore, because the microdomain was interrupted at the interface of the P3HT nanofibers, many P3HT nanofibers observed in the SFM image were embedded in the matrix. In addition, the P3HT molecules, which did not participate in the nanofiber formation, also scarcely affected the microphase separation of SBS because the domain spacing apart from the nanofibers was almost the same as that of the cast film of the pristine SBS solution.

From the magnified image of Figure 2b, no selective location of either component of SBS could be observed around the P3HT nanofiber. No specific component of the styrene or butadiene block was unevenly distributed around the interface between the SBS matrix and the nanofibers, which was embedded in a matrix for interrupting the microdomain. If there is a preferential interaction between a specific component and nanofibers, nanofibers should be surrounded by this component or aggregated each other. So, it is suggested that there is no preferential interaction between the P3HT nanofibers and each component of SBS.

### 3.2. FET Properties of SBS Film Loaded with a P3HT Nanofiber

Figure 3 shows the *I*-*V* characteristics of the thin film obtained by spin-casting the suspension of P3HT (0.05 wt%) and SBS (0.5–5 wt%) in the mixture of chloroform and anisole (70:30 (*v*/*v*)). On the polymer composite, because the P3HT molecules were exposed to a small amount of water or oxygen adsorbed in the matrix polymer, the film was difficult to be undoped completely even under the vacuum atmosphere. Therefore, the film had low conductivity even without doping. Increasing the ratio of SBS decreased conductivity, and then the conductivity was too small to be measured in the weight ratio of P3HT less than 0.99 wt% (P3HT: 0.05 wt%, SBS: 5 wt%). Generally, polymer composites loading anisotropic fillers with small diameters and high aspect ratios, such as CNTs, are reported to exhibit a low percolation threshold from 0.0021 to 8 wt% [49]. Furthermore, in the CNT/P3HT system, the percolation threshold was reported to be 1.0 wt% [50]. In the composite films loaded with P3HT, the percolation threshold was reported to be less than 1 wt% [31]. The small percolation threshold of our study was consistent with the previous studies. From the slope, the conductivity of SBS of 0.5 wt% was estimated to be 4.43 × 10^−4^ S cm^−1^.

We performed the FET measurements of the SBS films loaded with the P3HT nanofibers (P3HT: 0.05 wt%, SBS: 0.5 wt%). Figure 4a shows the output characteristics of the thin films by changing the gate voltage *V*_G_ from −20 V to −50 V. The drain current (*I*_SD_) vs. drain voltage (*V*_SD_) deviated from the Ohmic law to saturate at *V*_SD_~−10 V. Further, *I*_SD_ showed significant amplification with increasing negative *V*_G_, which is a typical property of p-type semiconductors. Figure 4b shows the transfer characteristic at *V*_SD_ = −30 V. As mentioned above, an apparent current could be detected at *V*_G_ = 0; therefore, the characteristics were regarded as normally on state. A typical p-type semiconductor property was observed, where the marked amplification of *I*_SD_ with respect to the negative *V*_G_ led to an *I*_SD_ of approximately −400 nA at a *V*_G_ of −50 V. Initially, *I*_SD_ increased in the low-negative-*V*_G_ region below −40 V; further, it tended to saturate. Here, *μ* was estimated to be 6.96 × 10^−3^ cm^2^ V^−1^ s^−1^, which was consistent with the previous studies on P3HT nanofibers composited in PMMA or PS [30,31,36]. Moreover, the on/off ratio was 4.90 × 10^3^. From the results of the electrical measurements, we found that the P3HT nanofiber network was connected in the SBS bulk matrix, and the SBS film loaded with the P3HT nanofiber had a p-type semiconducting property, which was expected to be applicable to the sheet transistor.

### 3.3. Electrical Properties of SBS Film Loaded with P3HT Nanofibers under Elongation and Bending Deformation

With increasing P3HT ratio, the conductivity increased but the mechanical strength decreased apparently. So, we measured the conductivity of the self-standing film containing P3HT (0.05 wt%) and SBS (5 wt%), which was the ratio obtaining self-standing film with a sufficient mechanical strength, after an elongation treatment. SFM was performed after the film was stretched by two times, and the SFM phase images of both the unstretched and stretched parts are shown in Figure 5. In the unstretched part, the P3HT nanofibers were observed frequently, while in the stretched part, these nanofibers were not observed at all. However, dark tracks, which were suggested to be the tracks of the nanofibers and existed before the elongation, could be observed. From this result, it was found that the P3HT nanofibers could not follow the elongation of the SBS matrix and were retained in the less-deformed part. As no specific component of the styrene or butadiene block was unevenly distributed around the interface between the SBS matrix and the nanofibers from the SFM images and no specific interaction was suggested to exist at the nanofiber interface, the results were consistent with this suggestion.

Figure 6a shows *I*-*V* characteristics of the self-standing film in the unstretched and stretched parts after the elongation. In the stretched part, the conductivity decreased by two orders of magnitude, and the conductive network was fatally broken. This result was consistent with the SFM observations. Although the SBS matrix has sufficient elasticity, the conductive nanofiber network could not follow this bulk elongation.

Figure 6b shows the conductance ratio of the stretched part *G* to the unstretched part *G*_0_ with changing a strain *ε*. In the SBS film loaded with P3HT nanofibers, the conductance decreased apparently at *ε* > 0.2, and the conductive property was no longer observed at *ε* = 1.0 (stretching by two times). In the previous study, the electrical property was maintained at *ε* of 0.5–1.0 [43,44,45,46], therefore the SBS and P3HT nanofibers had low resistance to the elongation deformation. From this result, it was found that the modification of the interface between each component of SBS and the P3HT nanofiber should be needed.

For the FET measurements of the bending deformation, a flexible film comprising the P3HT nanofibers (0.05 wt%) and SBS (5 wt%) on a PP substrate with a top gate was fabricated. The flexible SBS film loaded with the P3HT nanofibers with the electrodes was bent along a cylindrical mold with a radius of curvature of 1.75 cm, as shown in Figure 7.

Figure 8 shows the transfer characteristics of the flexible SBS film loaded with the P3HT nanofibers before and after the bending deformation at *V*_SD_ = −30 V. An apparent current could be detected at *V*_G_ = 0. Furthermore, the amplification with respect to negative *V*_G_ was scarcely observed in this case, whereas *I*_SD_ was depressed by positive *V*_G_. In this case, impurities could not be removed from the film comprising various components, and we could not obtain an off-state and a proper gate-controlled channel. As a result, the device operated in a normally-on state during the FET operation.

The field effect mobility before and after the bending deformation was estimated to be 1.64 × 10^−5^ cm^2^ V^−1^ s^−1^ and 5.30 × 10^−5^ cm^2^ V^−1^ s^−1^, respectively, and the mobility at the returning state from the bending deformation was estimated to be 1.35 × 10^−5^ cm^2^ V^−1^ s^−1^. The value was a considerably small value reported previously, such as the PDMS stretchable composite loaded with P3HT (the order of 10^−2^ cm^2^ V^−1^ s^−1^) [46]. This small value suggests that an optimization of the gate and gate insulator was not sufficient. However, the FET operation, that is, the depression of the drain current by *V*_G_ was hardly affected by the bending deformation, and we can show the possibility of fabrication of flexible devices using the SBS film loaded with the P3HT nanofibers.

## 4. Conclusions

We investigated the electrical properties of SBS composite films loaded with P3HT nanofibers. The P3HT nanofibers were embedded in the fingerprint pattern of the microphase separation of SBS, as noticed by SFM observation. A nanofiber network was formed in the SBS matrix, and the conductivity and FET properties were observed. The field-effect mobility, *μ*, was estimated to be 6.96 × 10^−3^ cm^2^ V^−1^ s^−1^, which was consistent with the previous study on the P3HT nanofibers composited in PMMA or PS. This film was stretchable, but after elongation by two times, the nanofibers could not follow the elongation of the SBS matrix; hence, the conductivity decreased drastically. The FET of this film was operated by bending deformation with a radius of curvature of 1.75 cm, though we could not obtain an off-state and the device operated in a normally-on state. The elastomer-based semiconducting sheet was one of the candidates for elastic and flexible semiconductors, which have the possibility for applying to biosensors as well as energy devices such as the electrodes of a flexible battery and capacitor.

## Figures and Tables

**Figure 1 polymers-12-02118-f001:**
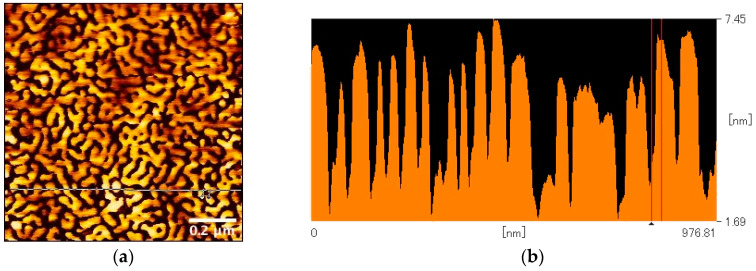
(**a**) Topographic image (1 × 1 µm) and (**b**) section plot of scanning force microscope (SFM) of the thin film by spin-casting the solution of poly(styrene-*b*-butadiene-*b*-styrene) (SBS) (0.5 wt%) in a mixture of chloroform and anisole (70:30 (*v*/*v*)). The distance between the red lines in the section plot corresponding to the half-domain spacing is 24.8 nm.

**Figure 2 polymers-12-02118-f002:**
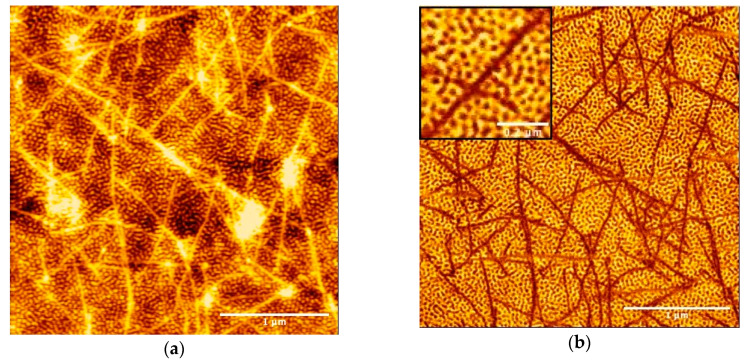
(**a**) SFM topographic and (**b**) phase images of the thin film obtained by spin-casting the suspension of poly(3-hexylthiophene) (P3HT) (0.05 wt%) and SBS (0.5 wt%) in the mixture of chloroform and anisole (70:30 (*v*/*v*)) (3 × 3 µm). In both the images, the microphase separation of SBS and the nanofiber formation of P3HT could be observed. Top left inset of (**b**) shows the magnified image of the phase image.

**Figure 3 polymers-12-02118-f003:**
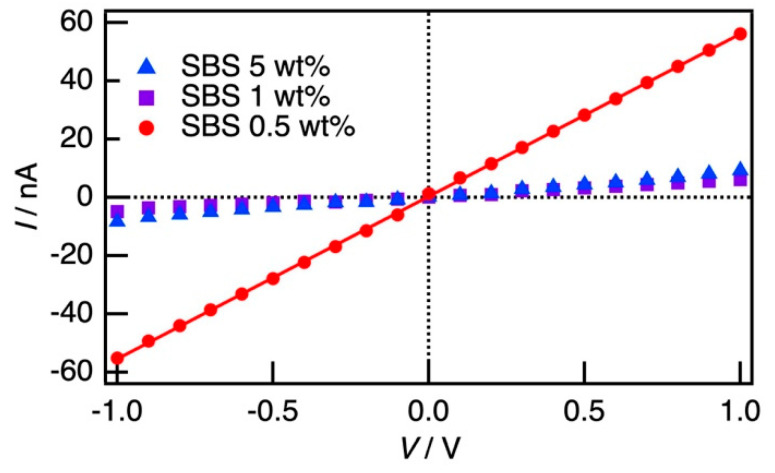
*I*-*V* characteristics of the SBS thin film loaded with the P3HT nanofiber.

**Figure 4 polymers-12-02118-f004:**
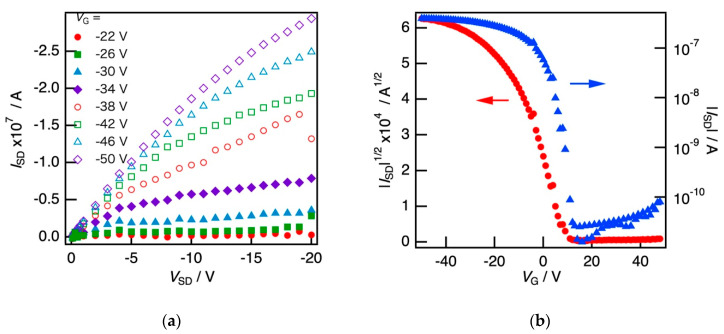
(**a**) Output characteristics and (**b**) transfer characteristics of the composite films of the SBS thin film loaded with the P3HT nanofibers.

**Figure 5 polymers-12-02118-f005:**
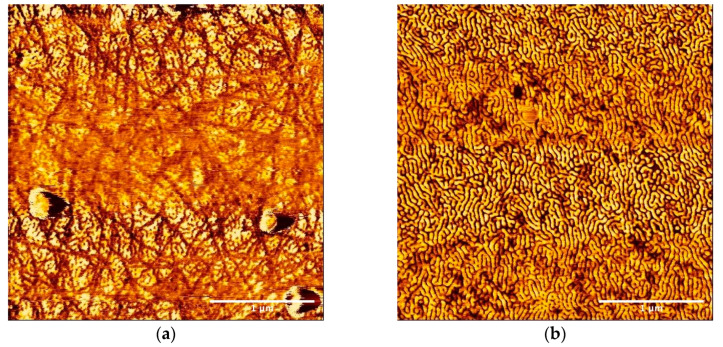
SFM phase image of the thin film obtained by spin-casting the suspension of P3HT (0.05 wt%) and SBS (0.5 wt%) in the mixture of chloroform and anisole (70:30 (*v*/*v*)) (3 × 3 µm) in (**a**) unstretched and (**b**) stretched parts after the elongation treatment.

**Figure 6 polymers-12-02118-f006:**
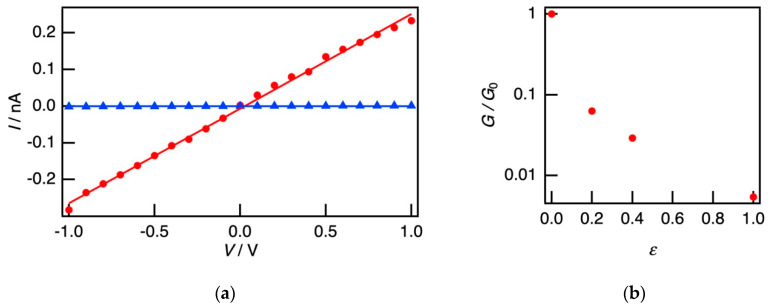
(**a**) *I*-*V* characteristics of the self-standing SBS film loaded with the P3HT nanofiber in the unstretched (red circles) and stretched (blue triangles) parts after the elongation treatment. (**b**) The ratio of the conductance of the stretched part *G* to the conductance of the unstretched part *G*_0_ with changing the strain *ε*.

**Figure 7 polymers-12-02118-f007:**
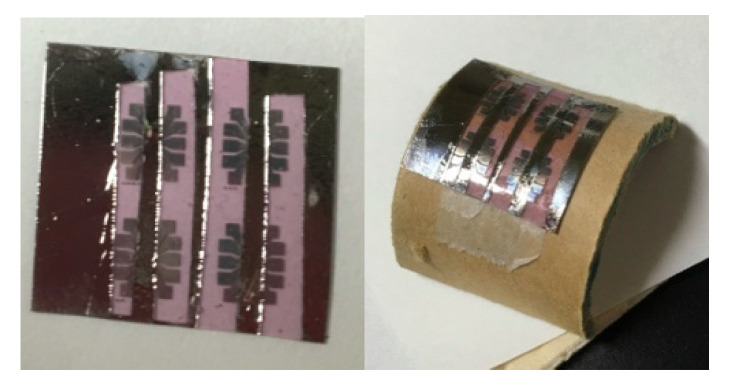
Photographic images of the flexible SBS film loaded with the P3HT nanofibers (**left**), and the film bent along the cylindrical mold with a radius of curvature of 1.75 cm (**right**).

**Figure 8 polymers-12-02118-f008:**
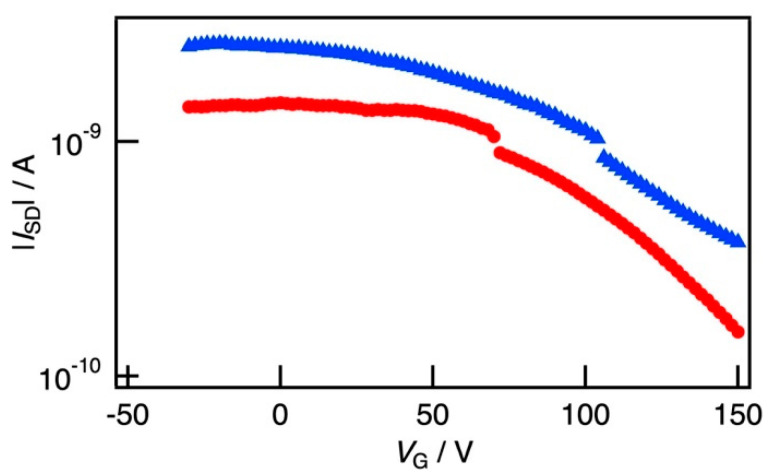
Transfer characteristics of the composite films of the thin SBS film loaded with the P3HT nanofiber before (red circles) and after (blue triangles) the bending deformation.

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
