# Peer review of "Semiconducting Properties of the Hybrid Film of Elastic Poly(styrene-b-butadiene-b-styrene) Block Copolymer and Semiconducting Poly(3-hexylthiophene) Nanofibers"

_polymers, 2020, doi:10.3390/polym12092118_

Round 1

Reviewer 1 Report

This manuscript investigated the electrical properties of a composite film loaded with semi-conductive 11 poly(3-hexylthiophene) (P3HT) nanofibers dispersed in poly(styrene-b-butadiene-b-styrene) (SBS). It's interesting to be one of the candidates for flexible semiconductors in molecular electronics. I suggest that more relevant experimental data can be perfected to more fully prove the importance, innovation, and possible contribution to the future of this work. The details as follows:

  1. Although we know that SBS is an excellent elastic resin, why did you choose it instead of other elastic resins (such as its replasticization, etc.)? And the molecular weight of SBS resin will directly affect its mechanical properties, why choose this molecular weight ?
  2. How do you determine the ratio of P3HT (0.05 wt%) and SBS (0.5 wt%) in this work? The ratio of different raw materials directly affects the conductivity and mechanical properties of the composite film, so we should study the relationship between the two through experiments. In other words, there is a lack of many comparative experiments. Maybe, the following literautre, e.g. PSHT in ‘Polymer 48(6) 1667-78’ and PANI in 'ACS Appl Mater Interfaces 12(5) 5820-30', are helpful for the above issue.
  3. In Figure 5, the experiment should be supplemented to further illustrate the relationship between different tensile forces and the distribution of P3HT, as well as the changes in film conductivity under different tensile forces in Figure 6. Express it scientifically with diagrams.
  4. In Figure 7 .What is the adaptable bending radius of curvature for the flexible SBS of P3HT nanofibers and electrodes? In other words, why did you choose 1.75cm?

Author Response

Answers to the reviewers’ comments

Thank you for your letter and for the reviewers’ comments concerning our manuscript entitled “Semiconducting Properties of the Hybrid film of Elastic Poly(styrene-b-butadiene-b-styrene) Block Copolymer and Semiconducting Poly(3-hexylthiophene) Nanofibers (title has been changed according to the comment of a scientific editor)” (ID: polymers-924807). Those comments are all valuable and very helpful for revising and improving our paper, as well as the important guiding significance to our researches. We have studied comments carefully and have made correction which we hope meet with approval.

Reviewer #1:

This manuscript investigated the electrical properties of a composite film loaded with semi-conductive 11 poly(3-hexylthiophene) (P3HT) nanofibers dispersed in poly(styrene-b-butadiene-b-styrene) (SBS). It's interesting to be one of the candidates for flexible semiconductors in molecular electronics. I suggest that more relevant experimental data can be perfected to more fully prove the importance, innovation, and possible contribution to the future of this work. The details as follows:

1) Although we know that SBS is an excellent elastic resin, why did you choose it instead of other elastic resins (such as its replasticization, etc.)? And the molecular weight of SBS resin will directly affect its mechanical properties, why choose this molecular weight?

[Reply] SBS is one of the suitable block copolymers for observing the phase structure by the scanning force microscope because it has a rigid polystyrene block and an elastic polybutadiene block. In this study, as we would like to investigate the P3HT nanofiber formation in the microphase separation by the scanning force microscope, SBS was selected.

As reviewer’s comment, the molecular weight was important factor for mechanical properties. In this point, we selected SBS with comparably large molecular weight and with fixed component ratio. By using polymers with smaller molecular weight, it is difficult to observe a lamellar phase of micro-phase separation. While by using polymers with larger polymerization, a polydispersity increases and a copolymerization ratio is incorrect, then the structure of the micro-phase separation becomes into disorder.

We added the following sentences in introduction.

[Add] (p2, L63) Furthermore, as SBS is one of the suitable block copolymers for observing the phase structure by the scanning force microscope (SFM), we can confirm the formation and dispersion of P3HT nanofibers in SBS matrix by the SFM observation.

2) How do you determine the ratio of P3HT (0.05 wt%) and SBS (0.5 wt%) in this work? The ratio of different raw materials directly affects the conductivity and mechanical properties of the composite film, so we should study the relationship between the two through experiments. In other words, there is a lack of many comparative experiments. Maybe, the following literautre, e.g. PSHT in ‘Polymer 48(6) 1667-78’ and PANI in 'ACS Appl Mater Interfaces 12(5) 5820-30', are helpful for the above issue.

[Reply] We added the I-V profiles of the composites with other concentration in Fig.2 for clarifying the reason of fixing the P3HT/SBS ratio. Furthermore, we added and changed some sentences and performed the comparison of the percolation threshold with comparative reports.

[Add] (p4, L165) With increasing the ratio of SBS the conductivity decreased, and then the conductivity too small to be measured in the weight ratio of P3HT less than 0.99 wt% (P3HT: 0.05 wt%, SBS: 5 wt%). Generally, polymer composites loading anisotropic fillers with small diameters and high aspect ratios, such as CNTs, are reported to exhibit a low percolation threshold from 0.0021 to 8 wt% [43]. Furthermore, in the CNT/P3HT system, the percolation threshold was reported to be 1.0 wt% [44]. In the composite films loaded with P3HT, the percolation threshold was reported to be less than 1 wt% [31]. The small percolation threshold of our study was consistent with the previous studies.

[Change] (p6, L207) With increasing P3HT ratio, the conductivity increased but the mechanical strength decreases apparently. So, we measured the conductivity of the self-standing film containing P3HT (0.05 wt%) and SBS (5 wt%), which was the ratio obtaining self-standing film with a sufficient mechanical strength, after an elongation treatment.

3) In Figure 5, the experiment should be supplemented to further illustrate the relationship between different tensile forces and the distribution of P3HT, as well as the changes in film conductivity under different tensile forces in Figure 6. Express it scientifically with diagrams.

[Reply] We added Figure 6b showing the conductance with changing the strain, and the comparison with the previous report. SFM images under small strain are difficult to be measured because stress is unevenly applied in a microscopic scale. In Figure 5, we intended to show the structure under sufficiently large elongation.

[Add] (p7, L243) Figure 6b shows the ratio of the conductance of the stretched part G to the conductance of the unstretched part G0 with changing a strain ε. In the SBS film loaded with P3HT nanofibers, the conductance decreased apparently at ε > 0.2, and the conductive property was no longer observed at ε = 1.0 (stretching by two times). In the previous study, the electrical property was maintained at ε of 0.5–1.0 [43–46], therefore the SBS and P3HT nanofibers had low resistance to the elongation deformation. From this result, it was found that the modification of the interface between each component of SBS and the P3HT nanofiber should be needed.

4) In Figure 7. What is the adaptable bending radius of curvature for the flexible SBS of P3HT nanofibers and electrodes? In other words, why did you choose 1.75cm?

[Reply] By our system using the probe station, we could not measure the small curvature less than 1.75 cm. In the small curvature, it was difficult to contact the probes to the source, drain, and gate electrodes in a reliable way.

Reviewer 2 Report

A manuscript contain original research work, there the authors investigated the electrical properties (electrical conductivity and field-effect mobility) of a composite film based on poly(styrene-b-butadiene-b-styrene) (SBS) loaded with poly(3-hexylthiophene) nanofibers dispersed in SBS. Proposed new materials could have the potential in the application as flexible semi-conductors in different electronic devices. I recomend to accept current manuscript, however there are some aspects in this manuscript that should be improved, major revision required:

  1. The novelty of the work is not highlighted in the manuscript, thereby the readers could not valuably evaluate the work.
  2. The discussion of obtained results missed the comparison of new composite materials with others.
  3. The application area also not enough demonstrated.
  4. Introduction section provides good impression, the authors clearly described the research objects and introduced new material. Only one drawback was found, pages 1-2, lines 25-63, authors describe novelty and importance of research team, however cited references are mainly before 2010 year. Only 20% references are from 2011-2019 years and only  three of them are from last five years. Thereby I highly recomend to renew a reference list by adding the references of 2016-2020 years.
  5. The authors got an interesting result:"nanofiber scarcely affected the microphase separation of SBS". It is necessary to offer the explanation of this phenomenon. How do the authors explane the fact that the distance between the crystal domains is preserved when embedding conductive fibers?
  6. Why in fig. 4b there are two blue lines in the region of Vg=10-50 V?
  7. In the experimental part, after equation (1), line 104, authors must add  the notation of terms L, W, Isp, Vg.

Author Response

Answers to the reviewers’ comments

Thank you for your letter and for the reviewers’ comments concerning our manuscript entitled “Semiconducting Properties of the Hybrid film of Elastic Poly(styrene-b-butadiene-b-styrene) Block Copolymer and Semiconducting Poly(3-hexylthiophene) Nanofibers (title has been changed according to the comment of a scientific editor)” (ID: polymers-924807). Those comments are all valuable and very helpful for revising and improving our paper, as well as the important guiding significance to our researches. We have studied comments carefully and have made correction which we hope meet with approval.

Reviewer #2:

A manuscript contain original research work, there the authors investigated the electrical properties (electrical conductivity and field-effect mobility) of a composite film based on poly(styrene-b-butadiene-b-styrene) (SBS) loaded with poly(3-hexylthiophene) nanofibers dispersed in SBS. Proposed new materials could have the potential in the application as flexible semi-conductors in different electronic devices. I recomend to accept current manuscript, however there are some aspects in this manuscript that should be improved, major revision required:

1) The novelty of the work is not highlighted in the manuscript, thereby the readers could not valuably evaluate the work.

[Reply] We added the following sentences in introduction for clarifying the novelty of the work.

[Add] (p1, L71) The novelty of this work is to develop the elastic and semiconducting polymer sheet using conducting polymers with a high elasticity as substitute for the rigid Si wafer. Although poly(3,4-ethyrene dioxythiophene):poly(styrenesulfonate) (PEDOT:PSS) is known to have high conductivity and applied to various applications, PEDOT:PSS is hydrophilic polymer, which cannot dispersed in almost elastomer with hydrophobic properties. The P3HT nanofiber is a realistic option for achieving the elastomer loaded with conducting polymers. There are some reports on this strategy by using polystyrene-b-poly(ethylene-co-butylene)-b-polystyrene [43–45], polydimethylsiloxane (PDMS) [46,47], silicon rubber [48]. In this study, we particularly focused on the relation between the structure of the microphase separation and P3HT nanofibers, which has not been discussed in the previous study.

2) The discussion of obtained results missed the comparison of new composite materials with others.

[Reply] We performed the comparison with other composites on the percolation threshold, the electrical conduction under the elongation, and FET under bending deformation.

(On the percolation threshold)

[Add] (p4, L165) With increasing the ratio of SBS the conductivity decreased, and then the conductivity too small to be measured in the weight ratio of P3HT less than 0.99 wt% (P3HT: 0.05 wt%, SBS: 5 wt%). Generally, polymer composites loading anisotropic fillers with small diameters and high aspect ratios, such as CNTs, are reported to exhibit a low percolation threshold from 0.0021 to 8 wt% [43]. Furthermore, in the CNT/P3HT system, the percolation threshold was reported to be 1.0 wt% [44]. In the composite films loaded with P3HT, the percolation threshold was reported to be less than 1 wt% [31]. The small percolation threshold of our study was consistent with the previous studies.

[Change] (p6, L207) With increasing P3HT ratio, the conductivity increased but the mechanical strength decreases apparently. So, we measured the conductivity of the self-standing film containing P3HT (0.05 wt%) and SBS (5 wt%), which was the ratio obtaining self-standing film with a sufficient mechanical strength, after an elongation treatment.

(On the electrical conduction under the elongation)

[Add] (p7, L243) Figure 6b shows the ratio of the conductance of the stretched part G to the conductance of the unstretched part G0 with changing a strain ε. In the SBS film loaded with P3HT nanofibers, the conductance decreased apparently at ε > 0.2, and the conductive property was no longer observed at ε = 1.0 (stretching by two times). In the previous study, the electrical property was maintained at ε of 0.5–1.0 [43–46], therefore the SBS and P3HT nanofibers had low resistance to the elongation deformation. From this result, it was found that the modification of the interface between each component of SBS and the P3HT nanofiber should be needed.

(On FET under bending deformation)

[Add] (p7. L266) The field effect mobility before and after the bending deformation was estimated to be 1.64 × 10-5 cm2 V-1 s-1 and 5.30 × 10-5 cm2 V-1 s-1, respectively, and the mobility at the returning state from the bending deformation was estimated to be 1.35 × 10-5 cm2 V-1 s-1. The value was considerably small value reported previously such as the PDMS stretchable composite loaded with P3HT (the order of 10-2 cm2 V-1 s-1) [46]. This small value is suggested that an optimization of the gate and gate insulator was not sufficient.

3) The application area also not enough demonstrated.

[Reply] We added the discussion of application area. Please see reply 2).

4) Introduction section provides good impression, the authors clearly described the research objects and introduced new material. Only one drawback was found, pages 1-2, lines 25-63, authors describe novelty and importance of research team, however cited references are mainly before 2010 year. Only 20% references are from 2011-2019 years and only three of them are from last five years. Thereby I highly recomend to renew a reference list by adding the references of 2016-2020 years.

[Reply] We added 7 references newly reported.

5) The authors got an interesting result:"nanofiber scarcely affected the microphase separation of SBS". It is necessary to offer the explanation of this phenomenon. How do the authors explane the fact that the distance between the crystal domains is preserved when embedding conductive fibers?

[Reply] We added the magnified image of Figure 2, and rearranged the explanation of this phenomenon for clarifying in detail.

[Change and Add] (p4, L143) Furthermore, because the microdomain was interrupted at the interface of the P3HT nanofibers, many P3HT nanofibers observed in the SFM image were embedded in the matrix. In addition, the P3HT molecules, which did not participate in the nanofiber formation, also scarcely affected the microphase separation of SBS because the domain spacing apart from nanofibers was almost the same as that of the cast film of the pristine SBS solution.

From the magnified image of Figure 2b, no selective location of either component of SBS could be observed around the P3HT nanofiber. No specific component of the styrene or butadiene block was unevenly distributed around the interface between the SBS matrix and the nanofibers, which was embedded in a matrix for interrupting the microdomain. If there is a preferential interaction between a specific component and nanofibers, nanofibers should be surrounded by this component or aggregated each other. So, it is suggested that there is no preferential interaction between the P3HT nanofibers and each component of SBS.

6) Why in fig. 4b there are two blue lines in the region of Vg=10-50 V?

[Reply] The vertical axis of Fig. 4b was drawn as an absolute value of ISD, which is the usual expression of FET transfer characteristics. As the value of ISD was very small value in the region of VG=10-50 V (almost 0), sign of measurement values frequently changed. In this case, the chart looks like there are two lines in the legalism plot. Many previous reports have the same tendency at the transfer characteristics.

7) In the experimental part, after equation (1), line 104, authors must add the notation of terms L, W, Isp, Vg.

[Reply] We added corresponding notations.

[Change] (p3, L123) where L is the spacing between the electrodes, W is the width of the electrodes, and Cox (= 13.8 nm cm-2) is the capacitance of the insulation layer of SiO2 (255 nm thick). ISD is the drain current and VG is the gate voltage.

Round 2

Reviewer 2 Report

The authors have seriously revised the manuscript, made corrections and significant additions in accordance with reviewer comments. The article can be accepted. Editing the English language is necessary.